# Overlap between Central and Peripheral Transcriptomes in Parkinson’s Disease but Not Alzheimer’s Disease

**DOI:** 10.3390/ijms23095200

**Published:** 2022-05-06

**Authors:** Kosar Hooshmand, Glenda M. Halliday, Sandy S. Pineda, Greg T. Sutherland, Boris Guennewig

**Affiliations:** 1Brain and Mind Centre, Faculty of Medicine and Health, School of Medical Sciences, University of Sydney, Camperdown, NSW 2050, Australia; khoo6186@uni.sydney.edu.au (K.H.); glenda.halliday@sydney.edu.au (G.M.H.); sandy.pineda@sydney.edu.au (S.S.P.); 2Garvan-Weizmann Centre for Cellular Genomics, Garvan Institute of Medical Research, Darlinghurst, NSW 2010, Australia; 3Charles Perkins Centre, Faculty of Medicine and Health, School of Medical Sciences, University of Sydney, Camperdown, NSW 2050, Australia; g.sutherland@sydney.edu.au

**Keywords:** Alzheimer’s disease, Parkinson’s disease, RNA sequencing, machine learning (ML), Brodmann Area 9 (dorsolateral prefrontal cortex), blood biomarkers

## Abstract

Most neurodegenerative disorders take decades to develop, and their early detection is challenged by confounding non-pathological ageing processes. Therefore, the discovery of genes and molecular pathways in both peripheral and brain tissues that are highly predictive of disease evolution is necessary. To find genes that influence Alzheimer’s disease (AD) and Parkinson’s disease (PD) pathogenesis, human RNA-Seq transcriptomic data from Brodmann Area 9 (BA9) of the dorsolateral prefrontal cortex (DLPFC), whole blood (WB), and peripheral blood mononuclear cells (PBMC) were analysed using a combination of differential gene expression and a random forest-based machine learning algorithm. The results suggest that there is little overlap between PD and AD, and the AD brain signature is unique mainly compared to blood-based samples. Moreover, the AD-BA9 was characterised by changes in ‘nervous system development’ with Myocyte-specific enhancer factor 2C (*Mef2C*), encoding a transcription factor that induces microglia activation, a prominent feature. The peripheral AD transcriptome was associated with alterations in ‘viral process’, and *FYN*, which has been previously shown to link amyloid-beta and tau, was the prominent feature. However, in the absence of any overlap with the central transcriptome, it is unclear whether peripheral *FYN* levels reflect AD severity or progression. In PD, central and peripheral signatures are characterised by anomalies in ‘exocytosis’ and specific genes related to the SNARE complex, including Vesicle-associated membrane protein 2 (*VAMP2*), Syntaxin 1A (*STX1A*), and p21-activated kinase 1 (*PAK1*). This is consistent with our current understanding of the physiological role of alpha-synuclein and how alpha-synuclein oligomers compromise vesicle docking and neurotransmission. Overall, the results describe distinct disease-specific pathomechanisms, both within the brain and peripherally, for the two most common neurodegenerative disorders.

## 1. Introduction

Alzheimer’s disease (AD) and Parkinson’s disease (PD) are well-known neurodegenerative disorders (NDs) that share common pathological events. They correlate with changes in gene expression that occur before the onset of and during the progression of these diseases [1]. Brodmann Area 9 (BA9) of the dorsolateral prefrontal cortex (DLPFC) plays an essential role in cognitive, motor, and memory-related functions and is pathologically affected in both diseases. AD and PD both feature prion-like spread of pathology [1] such that at post mortem, in any particular case, there will be brain regions at different stages of the disease [2,3]. Notably, at post mortem, the BA9 has mild to moderate pathological changes with most neurons intact. In this sense, analysis of BA9 transcriptome profiles may provide insights related to early neurodegenerative mechanisms involved in AD and PD [1,2].

Given the relative inaccessibility of the brain during a person’s lifetime, peripheral biomarkers that accurately reflect the state and progression of brain diseases and the responses to potential treatments are regarded as the ‘holy grail’ in the clinical management of NDs. Such markers have remained particularly elusive, although there is current optimism for a phosphorylated form of tau (p-tau217) as a blood biomarker in AD [4]. Blood biomarkers associated with central neuronal system (CNS) pathology could reflect the influence of the same genetic variants peripherally, systemic effects of the disease processes, and despite the bidirectional nature of the blood-brain-barrier (BBB), leakage or active transport of brain-borne pathological entities. Specifically, the release of mRNAs from apoptotic neurons into the plasma has already been described [3,4]. Given the common pathogenic mechanisms in NDs within the CNS, it is possible that these might also be observed peripherally.

Supervised machine learning algorithms show great promise for the analysis of ‘big data’ in ND research towards the diagnosis, prognosis, and development of new therapies [5]. Random forest, a supervised machine learning approach, has shown essential advantages over other methodologies, including the ability to handle highly nonlinearly correlated data, robustness to noise, tuning simplicity, and achieving the highest ratio of self-consistent selections in its results [6,7]. Therefore, the current study specifically applied a random forest-based algorithm with the feature selection tool, Boruta [8], to explore the transcriptome of the BA9, whole blood (WB), and peripheral blood mononuclear cells (PBMC) in human AD, PD, and cognitively healthy controls (NC) using RNA-sequencing (RNA-seq) data held in the National Centre of Biotechnological Information’s Sequence Read Archive (SRA; https://www.ncbi.nlm.nih.gov/sra (accessed on 1 September 2018)).

In theory, large data dimensions add more information to the dataset, thereby improving the quality of the data. However, one of the biggest challenges with the majority of public datasets, including SRA, is that there are more features or transcripts = *p* observed than patients = *n* characterised. This problem is known as the “curse of dimensionality” and substantially impacts analysis accuracy and recall due to overfitting that can mistake small fluctuations for significant variance in the data, and lead to spurious findings [9]. Therefore, the current study accessed the largest sample sizes held in SRA and utilised production bioinformatics, high-performance computing, and streamlined tools, coupled with differential gene expression and machine learning approaches to reduce noises as much as possible to avoid unnecessary complexity in the inferred models and improve the algorithm’s efficiency. The findings allowed for a comprehensive view of the publicly available RNA-Seq data in AD/PD and provide some evidence of shared genes and molecular pathways driving NDs, but primarily show distinct disease-specific patho-mechanisms both within the brain and peripherally for these NDs.

## 2. Results

### 2.1. Non-Overlapping Sample Subset Selection Using Differentially Expressed Genes

Aiming to uncover the molecular reconfigurations underlying PD and AD, the two most common NDs, large-scale RNA-seq studies were accessed through the Sequence Read Archive (SRA) database (Table 1, Figure 1A and Appendix A). Since they are variations across specific study designs, a subset of non-overlapping samples from the BA9, whole blood (WB), and peripheral blood mononuclear cells (PBMC) in human AD, PD, and cognitively healthy controls were selected (Table 1 and Appendix A).

A differential analysis using the negative binomial distribution model and the exact test was performed. First trimming, quality control, and alignment of the reads to the GRCh38 reference genome for the SRA data were processed utilising a production bioinformatics pipeline (Figure 1B, https://github.com/binfnstats/SRA-RNAseq).

Then, the SRA-tissue/diseases specific data were combined and normalised together (Figure 1C). Principal components (Appendix A) and hierarchical clustering (Appendix A) were calculated for all samples. An ANOVA test was performed to find confounding covariates with RIN, Braak, age of death, gender and project-IDs (studies) included in the model (Appendix A). Differentially expressed genes (DEGs) were identified for all matched independent subtypes, comparing the subtypes to the other samples (Table 2 and Appendix A). Functional enrichment of the DEGs was also tested to identify GO terms associated with each of the subtypes (Table 2 and Appendix A). Accordingly, combined independent studies with GO terms that showed clear relation to the disease/tissue of interest were established as the final selected subsets.

This was followed by the application of random-forest-based machine learning (ML) algorithm with the feature selection tool Boruta (Figure 1D), to identify novel diagnostic and prognostic markers and therapeutic targets in both peripheral and brain tissues of NDs.

### 2.2. ML-Based Ranking of Genes in BA9 (or DLPFC) from AD and PD

At autopsy, the moderately affected BA9 (or DLPFC) in AD and Lewy body disorders such as PD may harbour early pathogenic clues to pathomechanisms common to both diseases. This hypothesis was tested by re-analysing RNA-seq data using the Boruta feature selection method. There were 180 features (genes) that characterised AD BA9 samples (*n* = 101) compared to controls (*n* = 69) (Appendix A). The top-ranked Ontology (GO) biological processes (BPs) and in-common genes were ‘nervous system development’ (Solute carrier family 1 member 2 (*SLC1A2*), Cysteine rich transmembrane BMP regulator 1 (*CRIM1*), Myelin associated oligodendrocyte basic protein (*MOBP*), glycoprotein M6A (*GPM6A*), and Myocyte enhancer factor 2 (*MEF2C*)), and ‘mitochondrial transport’ (Heat shock protein 90 alpha family class A member 1 (*HSP90AA1*)) (Appendix A). Additionally, a connection between *MEF2C* and *GPM6A* was revealed through genetic interaction and/or co-expression analysis (Appendix A).

There were 189 genes that characterised PD DLPFC samples (*n* = 126) versus controls (*n* = 93) (Appendix A). The top PD-associated GO-BPs were ‘regulation of cellular response to heat’ (BAG cochaperone 3 (*BAG3*), Cysteine and histidine rich Domain containing 1 (*CHORDC1*), DnaJ heat shock protein family (Hsp40) member B1 (*DNAJB1*), Heat shock protein family A (Hsp70) member 8 (*HSPA8*), Heat shock protein family A (Hsp70) member *1A (HSPA1A*), and Heat shock protein family A (Hsp70) member 1B (*HSPA1B*)), ‘exocytosis’ (P21 (RAC1) activated kinase 1 (*PAK1*), Vesicle-associated membrane protein 2 (*VAMP2*), Syntaxin 1A (*STX1A*)), and ‘regulation of cellular response to stress’ (Solute carrier family 38 member 2 (*SLC38A2*)) (Appendix A). A connection was shown between *VAMP2*, and *STX1A* with synaptotagmin 1 (*SYT1*) (Appendix A). There were 235 distinct genes between AD compared to PD BA9 samples (Appendix A). The top GO-BPs were ‘intracellular protein transport (Transmembrane P24 trafficking protein 7 (*TMED7*))’ and ‘viral process’. Nucleophosmin 1 (*NPM1*), B Cell receptor associated protein 31 (*BCAP31*), and RAN binding protein 1 (*RANBP1*) were common in these two BPs (Appendix A). A connection was reported between *TMED7* and ADP ribosylation factor like GTPase 1 (*Arl1*) (Appendix A).

### 2.3. Diagnosis of AD and PD by Profiling Peripheral Blood Biomarkers Using ML

AD and PD-specific transcriptomic signatures were then obtained from PBMC and WB and analysed to determine if there was a feature overlap with the BA9 transcriptome for these two NDs. Fifty-three genes characterised the AD PBMC samples (*n* = 22) when compared to controls (*n* = 14) (Appendix A). Top GO-BPs were ‘cytoskeleton-dependent intracellular transport’ (Tubulin alpha 1a (*TUBA1A*), and Kinesin family member 1A (*KIF1A*)) followed by ‘viral process’ (Golgi brefeldin a resistant guanine nucleotide exchange factor 1 (*GBF1*), Growth factor receptor bound protein 2 (*GRB2*), Ankyrin repeat domain 17 (*ANKRD17*), S-phase kinase associated protein 1 (*SKP1), IK*, and *FYN*) (Appendix A). A connection was shown between *FYN* and Zeta chain of T cell receptor associated protein kinase 70 (*ZAP70*) (Appendix A). Seventy-three genes characterised PD PBMC samples (*n* = 6) versus controls (*n* = 11) (Appendix A). Top GO-BPs were ‘mRNA splicing, via spliceosome’ (Serine and arginine rich splicing factor 2 (*SRSF2*), Heterogeneous nuclear ribonucleoprotein U-like protein 1 (*HNRNPUL1*), *IK*, and Splicing factor proline and glutamine rich (*SFPQ*)) (Appendix A). There was a connection between *SFPQ* and Matrin 3 (*MATR3*) (Appendix A). There were 548 genes that differed between AD PBMC samples (*n* = 22) and PD (*n* = 6) (Appendix A). The top GO-BP was ‘immune system process’ (Major histocompatibility complex, class I, E (*HLA-E*), Major histocompatibility complex, class I, B (*HLA-B*), Major histocompatibility complex, class II, DR alpha (*HLA-DRA*), Major histocompatibility complex, class II, DR beta 1 (*HLA-DRB1*), Major histocompatibility complex, class II, DP alpha 1 (*HLA-DPA1*)) (Appendix A). Among all immune system related genes, *HLA-B* was strongly connected with killer cell immunoglobulin like receptor, three Ig domains and short cytoplasmic tail 1 (*KIR3DS1*) (Appendix A).

Among WB samples, 71 genes differed between AD (*n* = 48) and control samples (*n* = 22) (Appendix A). The top GO-BP was ‘interneuron migration from the subpallium to the cortex’ (ADP ribosylation factor like GTPase 13B (*ARL13B*)) (Appendix A). *ARL13B* and Tetratricopeptide repeat domain 26 (*TTC26*) were among the most prominent interacting genes (Appendix A).

A total of 180 genes differed between PD WB samples (*n* = 20) compared to controls (*n* = 20) (Appendix A). The top GO-BPs and genes were ‘exocytosis’ (*VAMP2*, *STX1A, PAK1*), as described above for PD-BA9 above and ‘regulation of cellular response to heat’ (MAPK activated protein kinase 2 (*MAPKAPK2*), *HSPA8*, *DNAJB1*, Glycogen synthase kinase 3 Beta (*GSK3B*), *HSPA1A*, *HSPA1B*, Calcium/Calmodulin dependent protein kinase II beta (*CAMK2B*), DnaJ heat shock protein family (Hsp40) member B6 (*DNAJB6*)) (Appendix A). Gene interaction and co-expression analysis were as seen in the PD-Ctrl BA9 comparison (Appendix A). Lastly, there were 84 genes that differed between AD WB samples (*n* = 48) compared to PD (*n* = 20) (Appendix A). The top GO-BP was ‘immune system process’ (*HLA-A*, *HLA-B*, and phosphatidylinositol-4,5-bisphosphate 3-kinase catalytic subunit delta (*PIK3CD*)) (Appendix A). This was followed by the same gene-interaction and co-expression results as seen in the PD-AD PBMC comparison (Appendix A).

### 2.4. Correspondence between Peripheral Blood and Brain in NDs

A direct comparison of blood and brain tissues for both diseases was also carried out.

There were 149 genes in common between PD-BA9 and WB samples (Figure 2(A1,A2)), with the top GO-BPs being ‘cellular response to heat’ (*HSPA1A*, and *HSPA1B*), ‘response to stress’, and ‘exocytosis’ (*PAK1*, *VAMP2*, and *STX1A*) (Figure 3A). Gene–gene interactions were as seen in the PD-Ctrl BA9 comparison (Figure 3B).

There were no DE genes in common between DLPFC and peripheral tissues in AD studies (Figure 3C).

## 3. Discussion

This study explored two overlapping hypotheses that there were common transcriptomic anomalies between AD and PD, and components of the central transcriptomic signatures would also be seen peripherally. The results suggest that there is little overlap between PD and AD centrally or peripherally and that the AD brain signature is essentially unique compared to blood-derived samples. In comparison, central and ‘blood’ PD signatures are both characterised by anomalies in ‘exocytosis’ with VAMP2, STX1A, and PAK1 the most prominent genes. There is some evidence too that PD is further characterised by aberrations in immune processes.

*VAMP2* and *STX1A*, critical proteins in the neuronal SNARE complex, were the prominent genes in the PD-BA9. *VAMP2*, a vesicle-associated (v)-SNARE, plays a crucial role in Ca^+2^-dependent exocytosis of synaptic vesicles, while *STX1A*, associated with the plasma membrane ((t)-SNARE), is specifically involved in vesicle fusion [10]. This appears consistent with a converging picture of the physiological role of α-synuclein (α-syn) as a modulator of SNARE complex assembly [11]. Pathologically, studies in mice have revealed an interaction of large α-syn oligomers with the N-terminus of multiple VAMP2s on vesicles. This incapacitates v-SNARE’s ability to interact with t-SNARE, inhibiting SNARE complex formation at the synaptic site and thus blocking exocytosis [12]. Interestingly, *VAMP2* and *STX1A* were co-expressed with *SYT1*. *SYT1* is a major Ca^2+^-sensor protein that regulates synaptic vesicle exocytosis through direct interactions with *VAMP2* and *STX1A* after Ca^2+^ influx in the presynaptic terminal [13]. Reduced levels of synaptic vesicle proteins (*SYT1, VAMP2*, and *STX1A*) along with increased expression of α-syn have been associated with synaptic dysfunction and neuronal degeneration in PD mice models [14].

A second finding here is that human leukocyte antigens, specifically the Class I molecules of the B gene (*HLA-B*), differentiate AD from PD blood-based transcriptomes. *HLA-B* significantly interacts with *KIR3DS1*. *KIR3DS*, a member of killer-immunoglobulin-like receptors (KIRs) is present on the surface of natural killer (NK) cells and can bind to *HLA-B* present on the surface of all nucleated cells. These interactions modulate NK cell activity, including the killing of virus-infected cells and tumours, or induction of cytokine secretion [15]. Changes in *HLA-B7* activity, along with a decrease in cytotoxic function of NK cells have been implicated as significant contributors to late-onset AD (LOAD) in subjects without apolipoprotein E (APOE) ε4 [16,17,18].

The most prominent feature of the AD- versus PD-BA9 comparison was *TMED7*, a type I transmembrane protein of the p24 protein family of the early secretory pathway, which is involved in the regulation of innate immune signalling. In a previous study, *TMED7* was shown to facilitate myeloid differentiation marker 88 (MyD88)-dependent toll-like receptor 4 (TLR4) signalling by forming a stable complex with its ectodomain and thus promoting TLR4 translocation from the Golgi to the cell surface through its vesicular trafficking activity [19]. Additionally, *TMED7* inhibits MyD88-independent TLR4 signalling by disrupting the TRIF-related adaptor molecule (TRAM)/TIR domain-containing adaptor protein inducing interferon β (TRIF) complex in late endosomes [20]. Although an association with NDs has not yet been established for *TMED7*, reduced levels of transmembrane protein 21KD (*TMP21*)/*TMED10*, a member of the p24 cargo receptor family, are known to be associated with AD progression [21,22]. Gene network analysis showed an interaction between *TMED7* and ADP-ribosylation factor-like protein 1 (*Arl1*), a member of the ARF/Arl family of small GTPases localised on trans-Golgi membranes [23]. Previous studies showed that depletion of *Arl1* causes dissociation of golgin-97, and golgin-245 is anchored to the trans-Golgi membrane via a carboxy-terminal GRIP domain, which binds to membrane-associated *Arl1* [23,24]. This impairs both vesicle trafficking and the Golgi integrity at the trans-Golgi network (TGN), which causes Golgi fragmentation, alteration of Golgi positioning, and impaired secretory traffic [23,25]. This is interesting since both *TMED7* and *Arl1* are involved in Golgi vesicle-mediated transport, and its alteration is likely to induce Golgi fragmentation in neurodegenerative diseases, including AD and PD [26].

*Mef2C*, a transcription factor implicated in the regulation of innate and adaptive immune cells with a myeloid origin, including microglia, was prominent in the AD-NC DLPFC comparison. In the CNS, *Mef2C* has been shown to restrict microglial responses to immune stimuli by functioning as an ‘off’ switch [27]. Studies on aged mice models have suggested that *Mef2C* loss of function mediates chronic elevation of type I interferon (*IFN-I*) response related to early microglial activation in AD-related conditions [27,28]. *Mef2C* is also involved in neuronal formation and differentiation, as well as in the growth and pruning of axons and dendrites through interaction with *GPM6A*. *GPM6A* is a member of the tetraspan proteolipid proteins (PLP) and a direct target gene for *Mef2C* that has been shown to be co-expressed with *Mef2C* [29].

Although not directly overlapping here, peripheral PBMC signatures did reveal genes that had previously been shown to be associated with AD in brain tissue analyses. Increased levels of *FYN* in AD brains have been proposed to regulate amyloid precursor protein (APP), phosphorylation at tyrosine 682 (Tyr682) residue, and be involved in oligomeric amyloid-β (Aβ)-mediated synaptic toxicity [30]. In turn, oligomeric Aβ increases local translation of the axonally-enriched protein Tau in the somatodendritic domain via activation of *FYN*/MAP kinase (*MAPK*)/ribosomal protein *S6* signalling pathway, thereby linking the two molecules that accumulate in AD brains [31]. Additionally, genetic interaction/co-expression network analysis revealed a connection between *FYN* and zeta-chain (TCR)-associated protein kinase, 70-kDa (*ZAP70*). Up-regulation of these two kinases is thought to increase peripheral mast cell and B-/T-cell activation, thus suggesting a role for peripheral blood immune dysregulation in AD [32,33].

A gene that specifically characterised the peripheral PD transcriptome encodes SFPQ, also known as polypyrimidine tract binding protein-associated-splicing factor (PSF), with roles in DNA transcription and repair and RNA processing. Previous PD animal model studies have reported dysregulated interactions between SFPQ/PSF and LIM homeobox transcription factor 1 beta (Lmx1b)/nuclear receptor-related 1 protein (Nurr1), which results in metabolic impairment, α-syn inclusions, and progressive loss of dopaminergic neurons [34,35]. Additionally, disrupted interactions between SFPQ/PSF and fused in sarcoma (FUS) has been shown in post mortem frontotemporal lobar degeneration (FTLD) brains and were associated with increased 4-repeat tau (4R-T) to 3-repeat tau (3R-T) ratios and underlying FTLD phenotype development. This seems likely due to dysregulated alternative splicing of microtubule-associated protein tau (MAPT) gene at exon 10 [36]. Additionally, our studies showed an interaction between *SFPQ* and *MATR3*, which is involved in DNA damage response (DDR) and activated by DNA double-strand breaks (DSB). *MATR3* and *SFPQ* depletion, along with increased levels of phospho-α-syn are associated with the accumulation of DNA DSBs that contribute to programmed cell death in in vitro models of PD [37,38].

One gene that defined AD-WB samples was *ARL13B*, a member of the Ras family of GTPases with distinct regulatory roles in primary cilia protein trafficking, the Sonic Hedgehog (Shh) pathway, and neural development [39]. Studies on AD mouse models have revealed mislocalisation of *ARL13B* on primary cilia membranes, a microtubule-based sensory organelle present in neurons and astrocytes. This led to defective cilia-mediated TLR4/NF-κB activation, reduced axonal length, and altered signal transmission [40,41]. Our analysis showed no genes interacting with *ARL13B*. However, *ARL13B* is co-expressed with *TTC26*. *TTC26*, also known as intraflagellar transport 56 (IFT56), which is responsible for *ARL13B* localisation on the cilia membranes. Reduced levels of *TTC26* lead to *ARL13B* mislocalisation and impaired Shh signalling pathway associated with abnormal maintenance of ciliary structure [42,43].

Overall, there is little overlap in either the central or peripheral transcriptomes of the two most common NDs, AD and PD. Unlike AD, there is an overlap between peripheral and central signatures of PD, suggesting anomalies in exocytosis and immune function. Thus, PD blood biomarkers may have more potential for monitoring the severity and progression of the disease. In the absence of blood biomarkers, the AD-BA9 signature reinforces the idea of anomalies in innate immune signalling and specifically TLR4/NF-κB activation.

## 4. Materials and Methods

### 4.1. Data Collection

To determine predictive NDs genetic variants and their biological function, we searched the ERA, Array Express and ENCODE databases along with the Sequence Read Archive (SRA) (https://www.ncbi.nlm.nih.gov/sra (accessed on 1 September 2018)) database (Figure 1A) to collate all publicly available RNA sequencing (RNAseq) studies derived from (*i*) human post mortem brain tissue and (*ii*) peripheral blood from patients with AD, PD, and healthy controls. The keywords utilised in the initial search were as follows: ‘Parkinson disease AND Prefrontal Cortex’, ‘Parkinson disease AND Blood’, ‘Alzheimer’s disease AND Prefrontal Cortex’, and ‘Alzheimer’s disease AND Blood’.

The initial search yielded 835 SRA fastq files, which recorded a total of 26 studies. Subsequently, all selected fastq files were downloaded using the getFASTQfile function from the SRAdb R/Bioconductor package [44]. All studies downloaded encompassed data from post mortem brain region: Brodmann area 9 only (BA9) (*n* = 155 with AD, *n* = 172 with PD, and *n* = 184 NC), and blood: PBMCs datasets (*n* = 33 with AD, *n* = 47 with PD, and *n* = 78 NC) and WB (*n* = 54 with AD, *n* = 38 with PD, and *n* = 74 NC) summarised in Table 1. Metadata obtained from each study were used to classify patients into NC, AD, and PD groups. In the SRA dataset, some of the AD and PD patients also had a definitive diagnosis according to Braak stage values, RNA integrity number (RIN), age of death, and gender (Appendix A).

### 4.2. Workflow Implementation and Data Processing

All the steps of the pipeline (https://github.com/binfnstats/SRA-RNAseq.git) were written in the workflow description language (WDL) that utilise Docker images, which run in Cromwell (Figure 1B). Using raw sequences as the input for the workflow, Trimmomatic (v0.38) [45] was applied to perform the quality trimming and filtering of reads with default settings. Then, FastQC (http://www.bioinformatics.babraham.ac.uk/projects/fastqc) was used to create a quality report for each pre-processed read file (https://github.com/binfnstats/SRA-RNAseq/blob/main/Scripts/Trim-%26_QC/Trim-QC.wdl). All trimmed RNA-Seq reads were aligned against the human genome version 38 (GRCh38) using the pseudoaligner Kallisto [46] with default settings (https://github.com/binfnstats/SRA-RNAseq/blob/main/Scripts/Kallisto/Kallisto.wdl). Kallisto was used as it strikes a balance between mapping runtime and similar or better accuracies compared with other alignment methods [47].

### 4.3. Transcriptomic Data Normalisation and Expression Analysis

Genes with low expression in all samples were removed using the “filterByExpr” function in the edgeR R/Bioconductor package [48], leaving 39,500 genes for downstream analysis. To correct any compositional bias from our data, Trimmed Mean of M-values (TMM) normalisation method [49] from “calcNormFactors” function in the edgeR package was used. Principal components analysis, hierarchical clustering, and ANOVA test were all performed and visualised using the R/Bioconductor package. Trimmed samples from different studies were randomly subdivided into groups using the “sample” function in R. All the differential expression analyses were performed in a negative binomial model-based method using edgeR. DE genes were filtered for those having the gene expression level Benjamini–Hochberg (BH) *Q* value < 0.05, and minimum fold change |logFC| ± 1) with the false-discovery rate (FDR) adjusted *p*-values ≤ 0.05.

### 4.4. Analysis of Genes Associated with NDs by Machine Learning (ML)

One of the biggest problems in a DE analysis in RNA-seq expression data is their high dimensionality and high pairwise correlations between genes. Some DEGs can be highly correlated and therefore contain irrelevant information, making it impractical to use all DEGs for developing diagnostic and prognostic prediction tools [50,51]. To address these problems and to further identify all relevant features in the datasets, we set to use the Boruta algorithm, a wrapper that is built around the random forest (RF) classifier [52,53]. Boruta-ML was run with max runs set to 20,000 for both classification and regression. The features which have importance significantly higher than the maximum Z score (MZS) were treated as relevant or confirmed.

### 4.5. Gene Ontology and Pathway Enrichment Analysis

To learn the potential functions of significant genes, topGO, an R/Bioconductor package [54], was used to perform Gene Ontology (GO) and Kyoto Encyclopedia of Genes and Genome (KEGG) enrichment analysis. Top-ranked GO biological pathways with an adjusted *p*-value of <0.05 were identified and used to select overrepresented or most enriched genes.

### 4.6. Gene–Gene Interaction and Gene Co-Expression Network Analysis

GeneMANIA webserver was used to identify genetic interaction (GI) and to represent co-expressed genes that are functionally coordinated with the selected gene biomarkers in response to a similar condition [55]. The study supports the better prediction of the most important functional genes that might provide a more robust bio-signature for phenotypic traits, thus providing more suitable biomarker candidates for future studies.

## 5. Conclusions

Applying a machine learning algorithm to existing central and peripheral transcriptomic has clarified that there is little overlap between the two diseases. The PD-DLPFC was characterised by alterations of SNARE proteins (VAMP2 and STX1A) which negatively affect neurotransmitter release at the axon terminal, while AD-BA9 was characterised by changes in ‘nervous system development’ and MEF2C, a transcription factor, potentially inducing microglia activation. Analyses of a gene signature from PBMC samples of AD subjects were associated with alterations in ‘viral process’ and FYN, which is associated with Aβ and tau in the brain but without overlap with the central transcriptome. It is unclear whether peripheral levels of FYN reflect disease severity and progression.

Whereas the central and peripheral PD-transcriptomes were both characterised by anomalies in exocytosis and particularly the differential expression of genes associated with the SNARE complex. This is consistent with our current understanding of the physiological role of α-syn and how oligomers may compromise vesicle docking and neurotransmission.

## 6. Potential Limitations and Future Prospective

Results from the present study have several notable limitations. Firstly, we could have missed relevant disease-associated gene sets across studies due to specific study designs, within-study biases, and variation across studies. Secondly, although we selected nearly 330 independent human peripheral RNA-seq studies for inclusion in our analyses, prioritising those most relevant to AD, PD, and cognitively healthy controls, we omitted many others with the potential to provide additional insights. In the future, our approach can thus be generalised to an even broader sample of available human peripheral data. Furthermore, through our analysis, a number of potentially essential genes were detected that remained incompletely defined and were likely to influence AD and PD. Therefore, it will be important in future work to consider gene sets significantly enriched for disease-specific SNPs through genome-wide association studies (GWASs). Moreover, human RNA-seq data derived from bulk brain tissue include mixed cell types composition, which may therefore reflect global changes in immunity and neuronal pathways. The growing availability of single-cell expression profiles can definitively address this concern in future work. Finally, as users of public data in which all cases are sporadic and do not include mutation carriers’ information, we expect heterogeneity, i.e., different patients having their combination of genetic and lifestyle factors. Still, this study is limited to finding common factors based on the assumption that all cases have a core of causative factors or a body of brain (expression) changes in common. The future work should include the hopefully available genotypes.

## Figures and Tables

**Figure 1 ijms-23-05200-f001:**
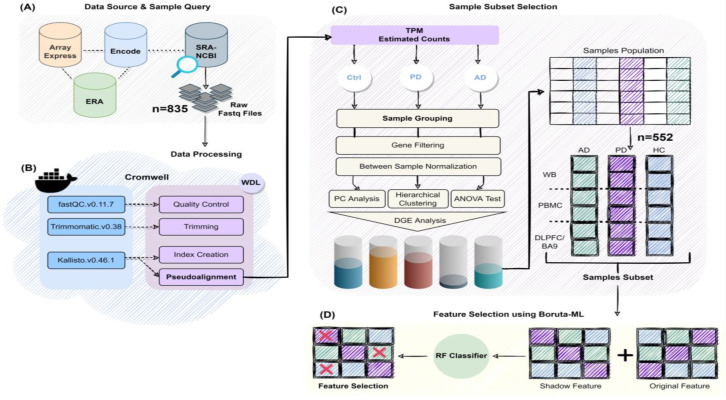
Schematic representation showing all steps necessary for RNA-seq analysis. (**A**) The Sequence Read Archive (SRA) database that makes an effort to collect the publicly available transcriptomics held in ERA, Array Express and ENCODE databases was searched for RNA-seq data to access publicly available human post mortem brain- and blood-based studies. (**B**) All the processing steps of the pipeline were wrapped into WDL tasks that were designed to be executed on the cloud-based services with Cromwell. Tasks in WDL workflow have an associated Docker-based tool image since WDL does not directly have the concept to build a tool. (**C**) Summary of the steps performed for the selection of a subset of samples using differential expression meta-analysis paradigm. (**D**) Boruta random forest (RF)-based algorithm was used as a feature selection method on normalised data. Boruta adds randomness to the given dataset by creating shuffled copies of all features, which are called shadow features. Then, it trains a random forest classifier on this extended data and applies a feature importance measure, and evaluates the importance of each feature.

**Figure 2 ijms-23-05200-f002:**
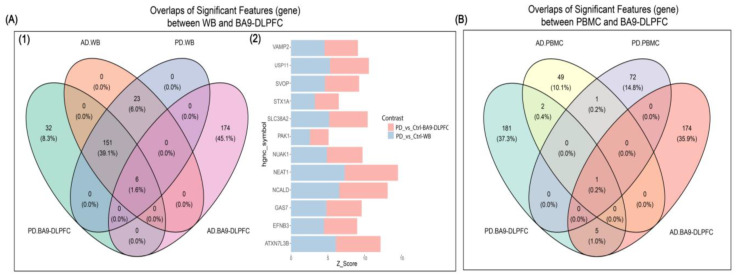
(**A**) (**1**) Venn diagrams illustrate the overlapping features (genes) evaluated with Boruta random forest (RF)-based algorithm between Brodmann Area 9 (BA9) of the dorsolateral prefrontal cortex (DLPFC) and whole blood (WB) samples of Parkinson’s disease (PD), Alzheimer’s disease (AD), and cognitively healthy controls (NC). In the digram, colors represent the followings; WB-AD-vs-Ctrl (orange), WB-PD-vs-Ctrl (blue), PFC-AD-vs-Ctrl (green), and PFC-AD-vs-Ctrl (purple). (**2**) The stacked plot reveals the top 12 significant features (based on their MZS) overlapped between brain (pink) and WB (blue) samples of PD patients. Vesicle-associated membrane protein 2 (VAMP2), ubiquitin specific peptidase 11 (USP11), SV2 related protein (SVOP), syntaxin 1A (STX1A), solute carrier family 38 member 2 (SLC38A2), P21 (RAC1) activated kinase 1 (PAK1), NUAK family kinase 1 (NUAK1), nuclear enriched abundant transcript 1 (NEAT1), neurocalcin delta (NCALD), growth arrest specific 7 (GAS7), ephrin B3 (EFNB3), ataxin 7 like 3 (ATXN7L38). (**B**) Venn diagrams illustrate the overlapping genes between BA9-DLPFC, and PBMC samples of PD, AD, and NC. In the digram, colors represent the followings; PBMC-AD-vs-Ctrl (yellow), PBMC-PD-vs-Ctrl (purple), PFC-AD-vs-Ctrl (orange), and PFC-AD-vs-Ctrl (green).

**Figure 3 ijms-23-05200-f003:**
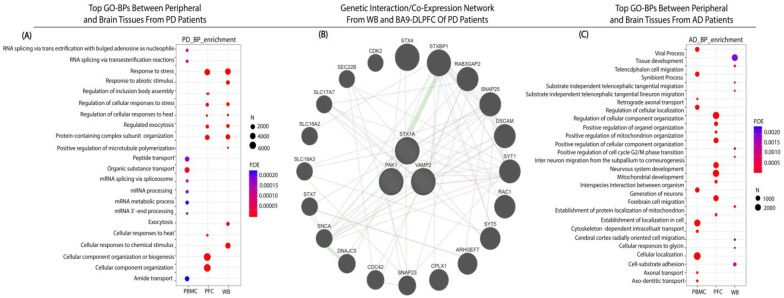
(**A**) Biological processes (BP) of Gene Ontology terms that were significantly enriched for overlapped features between BA9-DLPFC and WB samples of PD patients. In the plots, each dot’s color and size represent adjusted p.value (P.DE) and number of genes (N), respectively. (**B**) Gene interaction (green) and co-expression (blue) network of common significant genes between central and peripheral samples of PD patients. Syntaxin 1A (STX1A), P21 (RAC1) activated Kkinase 1 (PAK1), vesicle-associated membrane protein 2 (VAMP2). (**C**) BP of gene ontology terms between blood and brain tissue of AD patients. In the plots, each dot’s color and size represent adjusted p.value (P.DE) and number of genes (N), respectively.

**Table 1 ijms-23-05200-t001:** Number of tissue-specific samples per phenotype. For detailed information, please refer to Appendix A.

Condition	PBMC	WB	DLPFC/BA9	PBMC	WB	DLPFC/BA9
		Total Number of Samples			Final Number of Samples	
Parkinson’s Disease	47	38	172	6	20	126
Alzheimer’s Disease	33	54	155	22	48	101
Cognitively-Healthy Controls	78	74	184	25	42	162

**Table 2 ijms-23-05200-t002:** Differentially expressed genes (DEGs) and enriched GO-BPs from a selected subset of samples. For detailed information, please refer to Appendix A.

Tissue	Contrast	Number of DEGs	Top GO-BPs	Genes
DLPFC/BA9	AD-vs.-ctrl	8948	1-Intracellular transport 2-Cellular component organisation 3-Cellular protein localisation 4-Organelle organisation 5-Protein localisation 6-mRNA metabolic process 7-Peptide transport 8-Nitrogen compound transport	hondroitin sulfate proteoglycan 5 (CSPG5), DAAM1, SEPTIN9, kinesin family member 5C (KIF5C), Unc-51 like autophagy activating kinase 1 (ULK1), Ubiquitin-specific protease 9, X-LINKED (Usp9X), RAP2A, Transforming growth factor beta regulator 4 (TBRG4), TAP binding protein (TAPBP), Solute aarrier family 6 member 8 (SLC6A8)
DLPFC/BA9	PD-vs.-ctrl	12,043	1-Intracellular transport 2-Cellular protein localization 3-Cellular component organisation 4-Protein localisation 5-Intracellular protein transport 6-Cellular protein metabolic process 7-Catabolic process 8-mRNA metabolic process	RANBP1, Spire type actin nucleation factor 1 (SPIRE1), Solute carrier family 9 member A3 (SLC9A3), SEPTIN9, Peroxisomal biogenesis factor 10 (PEX10), VTI1B, Transmembrane protein 132A (TMEM132A), Trans-golgi network protein 2 (TGOLN2), Phosphotriesterase-related protein (PTER), Poly(RC) binding protein 2 (PCBP2)
DLPFC/BA9	AD-vs.-PD	8554	1-Intracellular transport 2-Cellular protein localisation 3-Protein localisation 4-Cellular component organisation 5-Intracellular protein transport 6-Organelle organisation 7-Peptide transport 8-Nitrogen compound transport	VPS41, SEPTIN9, ULK1, dishevelled associated activator of morphogenesis 1 (DAAM1), Dishevelled associated activator of morphogenesis 2 (DAAM2), Transmembrane P24 Trafficking Protein 7 (TME D7), Golgi reassembly-stacking protein 2 (GORASP2), TAPBP, SLC6A8
WB	AD-vs.-ctrl	740	1-SRP-dependent co-translational protein targeting to membrane 2-Nuclear-transcribed mRNA catabolic process, nonsense-mediated decay 3-Viral process 4-Viral transcription 5-Nuclear-transcribed mRNA catabolic process 6-Protein localisation to membrane	Ribosomal protein L31 (RPL31), Ribosomal protein L32 (RPL32), SMG5, H2AX, LSM4, Adaptor related protein complex 3 subunit delta 1 (AP3D1)
WB	PD-vs.-ctrl	5641	1-Granulocyte activation 2-Neutrophil degranulation 3-Neutrophil activation involved in immune response 4-Leukocyte degranulation 5-Neutrophil activation 6-Neutrophil mediated immunity 7-Myeloid leukocyte activation 8-Leukocyte activation involved in immune response	Vesicle associated membrane protein 8 (VAMP8), Myeloid differentiation primary response 88 (MYD88), Spleen associated tyrosine kinase (SYK), HCK, C-X-C Motif chemokine receptor 2 (CXCR2), WD repeat domain 1 (WDR1), Fc alpha receptor (FCAR), TYROBP, SYK
WB	AD-vs.-PD	3143	1-Neutrophil activation 2-Neutrophil activation involved in immune response 3-Neutrophil degranulation 4-Neutrophil mediated immunity 5-Myeloid leukocyte activation 6-Leukocyte activation 7-Regulated exocytosis 8-Vesicle-mediated transport	CXCR2, C-C motif chemokine ligand 5 (CCL5), Fc epsilon receptor Ig (FCER1G), TYRO protein tyrosine kinase binding protein (TYROBP), Stimulator of interferon response CGAMP interactor 1 (STING1), Major histocompatibility complex, class I, B (HLA-B), Major histocompatibility complex, class I, C (HLA-C), WDR-1, Peroxiredoxin 1 (PRDX1), PRDX2, Synaptogyrin 2 (SYNGR2), Myosin heavy chain 9 (MYH9), Reticulon 3 (RTN3), COPI coat complex subunit gamma 1 (COPG1), Perilipin 3 (PLIN3), ERGIC and Golgi 3 (ERGIC3)
PBMC	AD-vs.-ctrl	3921	1-mRNA metabolic process 2-Intracellular transport 3-Cellular protein localisation 4-Translational initiation 5-Cellular metabolic processes 6-Cotranslational protein targeting to membrane 7-Protein targeting to ER 8-SRP-dependent cotranslational protein targeting to membrane	Poly(RC) binding protein 2 (PCBP2), RNA binding protein 1 (RNABP1), SEPTIN9, ATP binding cassette subfamily E member 1 (ABCE1), Exosome component 10 (EXOSC10), SEC61 translocon subunit alpha 1 (SEC61A1), Translocation associated membrane protein 1 (TRAM1), Signal recognition particle 14 (SRP14), SRP receptor subunit alpha (SRPRA), Ribosomal protein L31 (RPL31), Ubiquitin A-52 residue ribosomal protein fusion product 1 (UBA52)
PBMC	PD-vs.-ctrl	8202	1-Immune system process 2-Viral process 3-Cellular metabolic process 4-cell activation 5-Immune response 6-Cell activation involved in immune response 7-Myeloid leukocyte activation 8-Leukocyte activation involved in immune response	Major histocompatibility complex, class II, DQ alpha 1 (HLA-DQA1), Major histocompatibility complex, class II, DR beta 1 (HLA-DRB1), Major histocompatibility complex, class I, F (HLA-F), HLA-C, Major histocompatibility complex, class I, E (HLA-E), Exosome component 1 (EXOSC1), TIMP metallopeptidase inhibitor 1 (TIMP-1), Golgi brefeldin a resistant Guanine nucleotide exchange factor 1 (GBF1), TYROBP, SYK
PBMC	AD-vs.-PD	6599	1-Viral process 2-Cellular metabolic process 3-Cellular component organisation 4-Intracellular transport 5-Cellular protein localisation 6-mRNA metabolic process 7-Nitrogen compound metabolic process 8-Immune system process	SPEN, Voltage dependent anion channel 1 (VDAC1), C-X-C motif chemokine receptor 4 (CXCR4), Exosome component 10 (EXOSC10), Formin Like 2 (FMNL2), RAB14, SEPTIN9, Poly(RC) binding protein 2 (PCBP2), Nitrilase 1 (NIT1), AT-Rich interaction domain 5B (ARID5B), DPA1, Leucine rich repeat containing G protein-coupled receptor 4 (LGR4), Ficolin 1 (FCN1), ETS proto-oncogene 1, transcription factor (ETS1), Macrophage expressed 1 (MPEG1)

## Data Availability

All data supporting the results are within the Manuscript and accompanying Figures or in Appendix A. All raw data used in this study were downloaded from the Sequence Read Archive (NCBI-SRA; https://www.ncbi.nlm.nih.gov/sra (accessed on 1 September 2018)); see Appendix A for all accessions.

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
