# Peer review of "Overlap between Central and Peripheral Transcriptomes in Parkinson’s Disease but Not Alzheimer’s Disease"

_ijms, 2022, doi:10.3390/ijms23095200_

Round 1
Reviewer 1 Report
ijms-1681420: A Comparative Analysis of the Central and Peripheral Transcriptomic Signatures of Alzheimer’s and Parkinson’s disease
This paper performed a so-called big data analysis of RNA-Seq transcriptomic data of the brain cortex, the whole blood, and peripheral blood mononuclear cells. It was found that there is little overlap between Parkinson’s disease (PD) and Alzheimer’s disease (AD). There is a rough believe that many neurodegenerative disorders should share the final neurodegenerative stages, although the initiation of the degeneration should be different from each other. Hence, the transcriptome of the AD and PD should include the common ones and specific ones. From this point, the present report, finding the complete difference between AD and PD, is very important and interesting. It is worth being published in the Journal. The manuscript is almost complete and can be published in the present form. Some points, which may improve the value of this review a little bit, are suggested as follows. Since the evaluation of the data processing software is very difficult, it is assumed that that the data analysis has been performed properly. This review is focusing on the biological matters.
(1) The heterogeneity of AD and PD, including their disease stages.
Since the objective biomarkers of AD or PD have not been established or just started being clinically used, the data of AD and PD should include may types of AD and PD, especially in the case of AD and non-AD type dementia (for example, https://pubmed.ncbi.nlm.nih.gov/35342092/, section "Inclusion of non-AD patients"). In addition, the transcriptome should change according to their disease stages. The qualification of the data base used in the present report and the heterogeneity of AD and PD patients should be briefly discussed in relation to the results and conclusion.
(2) Non alpha-synuclein PD
In relation to the above comment (1), PD can be divided into two subtypes, alpha-synuclein aggregation (Lewy bodies) positive and negative PD. In line 213, the matter of alpha-synuclein is discussed. However, some type of PD has little Lewy bodies (for example, PD with the mutation of parkin gene, https://www.ncbi.nlm.nih.gov/pmc/articles/PMC6046180/). This matter can be discussed briefly.
End of File
Author Response
Dear Ms Marcus
We sincerely thank your office and the two experts who reviewed our work. We are thankful for their valuable comments and suggestions, which have allowed us to significantly improve the manuscript. Please find below our responses to the Reviewers’ comments. We indicate how we addressed each comment and note the changes made to the text. Our answers are highlighted in blue and textual changes in the revised manuscript are marked up using ‘Track Changes’ function. We have also gone through the revised manuscript and corrected all other typos found.
Response to Reviewer 1 Comments
Point 1: The heterogeneity of AD and PD, including their disease stages.
Since the objective biomarkers of AD or PD have not been established or just started being clinically used, the data of AD and PD should include many types of AD and PD, especially in the case of AD and non-AD type dementia (for example, https://pubmed.ncbi.nlm.nih.gov/35342092/, section "Inclusion of non-AD patients"). In addition, the transcriptome should change according to their disease stages. The qualification of the data base used in the present report and the heterogeneity of AD and PD patients should be briefly discussed in relation to the results and conclusion.
Response 1:
For brain samples, this study collected pathologically confirmed cases; therefore, we excluded the non-AD dementias. To the best of our knowledge, all patients were sporadic and did not include mutation carriers. Nevertheless, we agree with Reviewer 1 that there will be heterogeneity as different cases will have their combination of genetic and lifestyle factors.
However, our study aimed to find common factors based on the assumption that all cases will have a core of causative factors and, therefore, some brain (expression) changes in common.
Please note that the above discussion has been summarised and added to the “Potential limitations and future prospective” section of the revised manuscript.
Point 2: Non alpha-synuclein PD.
In relation to the above comment (1), PD can be divided into two subtypes, alpha-synuclein aggregation (Lewy bodies) positive and negative PD. In line 213, the matter of alpha-synuclein is discussed. However, some type of PD has little Lewy bodies (for example, PD with the mutation of parkin gene, https://www.ncbi.nlm.nih.gov/pmc/articles/PMC6046180/). This matter can be discussed briefly.
Response 2:
We agree with Reviewer 1 that juvenile-onset PD resulting from parkin (PARK2) mutations do not show the typical Lewy pathology seen in other PD cases. Monogenic forms of AD/PD are rare, but as discussed above we assume that none of the cases are mutation carriers, but this is based on an absence of mutation screening in the original studies.
Reviewer 2 Report
In this manuscript Hooshmand and colleagues reanalyzed RNA-sequencing (RNA-seq) data from affected brain region (Brodmann Area 9, BA9) and blood cells of Alzheimer’s disease (AD) and Parkinson’s disease (PD) patients and healthy controls, in an attempt to assess whether there is an overlapping transcriptomic signature between AD and PD and more importantly between brain and blood cells. Using machine learning algorithm and complicated data analyses, the authors show that there is little overlap between PD and AD, and no overlapping between brain and blood in AD samples. Therefore, the data imply the difficulties to find a blood derived biomarker for AD diagnosis and progression prediction. The study is well designed with interesting results that is informative to the neurodegenerative research field. A few comments below are for authors consideration to address.
- The title of the manuscript needs to be more specific. At the time transcriptomic studies are wide spread across neuroscience field, “A comparative analysis..” is not that interesting. Why not tell the results, conclusion directly?
- If the authors are confident about the data that there is virtually no overlapping of genes between brain and blood samples to serve as AD biomarker candidates, the implication of the current research should be discussed more specifically about the perspectives of blood biomarker development, so the people in the field get the message more clearly, there is little chance of blood biomarker success for AD prediction.
- The sentence on line 72 “more features = p observed (transcripts, genes, etc.) than patients = n characterized” is confusing. Why are transcripts and genes called ‘features”? Should number of genes (usually in the many thousands) in studies be more than number of patients? Please clarify.
- Some data in the supplemental may be moved to the front to highlight the genes involved and associated pathways.
Author Response
Dear Ms Marcus
We sincerely thank your office and the two experts who reviewed our work. We are thankful for their valuable comments and suggestions, which have allowed us to significantly improve the manuscript. Please find below our responses to the Reviewers’ comments. We indicate how we addressed each comment and note the changes made to the text. Our answers are highlighted in blue and textual changes in the revised manuscript are marked up using ‘Track Changes’ function. We have also gone through the revised manuscript and corrected all other typos found.
Response to Reviewer 2 Comments`
Point 1: The title of the manuscript needs to be more specific. At the time transcriptomic studies are wide spread across neuroscience field, “A comparative analysis” is not that interesting. Why not tell the results, conclusion directly?
Response 1:
We replaced the previous title with “Overlap Between Central and Peripheral Transcriptomes in Parkinson's disease but not Alzheimer's disease”.
Point 2: If the authors are confident about the data that there is virtually no overlapping of genes between brain and blood samples to serve as AD biomarker candidates, the implication of the current research should be discussed more specifically about the perspectives of blood biomarker development, so the people in the field get the message more clearly, there is little chance of blood biomarker success for AD prediction.
Response 2:
The above discussion has been added to the “Potential limitations and future prospective” section of the revised manuscript.
Point 3: The sentence on line 72, “more features = p observed (transcripts, genes, etc.) than patients = n characterized”, is confusing. Why are transcripts and genes called ‘features”?
Response 3:
In machine learning and pattern recognition, a feature is defined as an individual measurable property or characteristic of a phenomenon. Therefore, the term “Feature” proposed in this research describes genes and transcripts that reflect the underlying pathomechanisms of AD and PD (initiation or progression.
Please note that the term, transcript, has been added to the revised manuscript in line 72 to clarify the word “Feature” for readers.
Point 4: Should number of genes (usually in the many thousands) in studies be more than number of patients? Please clarify.
Response 4:
In data analysis, large datasets with the so-called “large, small” problem (where p is the number of features and n is the number of samples) tend to be prone to overfitting that can mistake small fluctuations for significant variance in the data, and lead to spurious findings. Therefore, noise should be reduced as much as possible to avoid unnecessary complexity in the inferred models and improve the algorithm's efficiency.
Please note that reference 8 has been added to the revised manuscript for readers interested in learning more about this fact. Also, changes have been applied to the text in the revised manuscript.
Point 5: Some data in the supplemental may be moved to the front to highlight the genes involved and associated pathways.
Response 5:
Thank you for this suggestion. We have now mentioned more genes and associated pathways in Table 2 of the revised manuscript that were previously only in the supplementary table.